# Diversity-Oriented Synthesis (DOS) Towards Calcitriol Analogs with Sulfur-Containing Side Chains

**DOI:** 10.3390/ijms26136266

**Published:** 2025-06-28

**Authors:** Uxía Gómez-Bouzó, Generosa Gómez, Yagamare Fall

**Affiliations:** Departamento de Química Orgánica and Instituto de Investigación Sanitaria Galicia Sur (IISGS), Campus Lagoas Marcosende, Universidad de Vigo, 36310 Vigo, Spain; ugomez@uvigo.gal

**Keywords:** vitamin D, calcitriol, cancer, VDR, synthesis, clinical application

## Abstract

We report the synthesis of a very useful building block that could give pharmaceutical companies access to a series of novel calcitriol analogs, a library of compounds. This approach is called Diversity-Oriented Synthesis (DOS).

## 1. Introduction

1α, 25-dihydroxy vitamin D_3_ (1,25(OH)_2_D_3_; calcitriol)_,_ the hormonally active form of vitamin D_3_, has been shown to exhibit a wide variety of antitumor activities, but its therapeutic use is limited due to its severe calcemic side effects at the effective dose required for antitumor activity [1]. Accordingly, there is much interest in the design and synthesis of new calcitriol analogs displaying less toxicity [2]. The pharmacodynamic interaction pattern between VDR and calcitriol and its analogs has been extensively studied [3,4,5,6,7], with key pharmacophoric contacts involved in VDR-mediated bioactivity.

In recent decades, one of the most attractive objectives in the design of calcitriol analogs has been the introduction of a heteroatom into the side chain. Heteroatoms, such as sulfur and nitrogen, offer unique opportunities to modify the biological activity of these compounds.

One of the most notable modifications is the substitution of a carbon atom with a sulfur atom at the C-22 of the calcitriol side chain, resulting in compounds such as 22-thiacalcitriol (TCT, Figure 1). This modification is of particular interest due to bioisosterism, where the sulfur atom can mimic the structure and function of the original oxygen or carbon, but with subtle differences that can lead to improved biological properties.

The first analog to effectively suppress PTH without inducing hypercalcemic side effects was 22-oxacalcitriol (OCT, Figure 1). Studies in parathyroid cell cultures indicated that OCT was as potent as calcitriol in reducing PTH secretion, and it is currently used in the treatment of secondary hyperparathyroidism. As a continuation of the study of side chain modification with a heteroatom, Kubodera et al. [8] synthesized and evaluated TCT. Preliminary in vitro assays of this analog demonstrated decreased activity-inducing differentiation of the HL-60 human myeloid leukemia cell line.

In an in vitro study of the differentiation-inducing activity of HL-60 cells into macrophages using several thia derivatives, it was shown that every 20(*R*)-isomer exhibited more potent activity than its corresponding 20(*S*)-isomer. Among these synthesized analogs, the most potent, 20(*R*)-1α,25-dihydroxy-26,27-dimethyl-22-thiavitamin D_3_ **1** (Figure 2), was approximately 100,000 times as active as calcitriol.

Subsequently, in another study conducted by DeLuca et al. [9], it was shown that analog **2** (Figure 2) is 20 times more active than the natural hormone in terms of transcriptional activity and approximately 100 times more potent in osteoclast formation.

## 2. Results and Discussion

Given these excellent biological results of TCT, **1** and **2**, we designed a synthetic approach to a useful and versatile synthon that could lead to new calcitriol analogs with a sulfur atom at the C-22 side chain.

We anticipated that we could access target synthon **6** using ketone **3**. Our retrosynthetic analysis for **6** is depicted in Figure 1.

Accordingly, chiral allylic alcohol 4 was prepared as outlined in Figure 2, using a new methodology developed in our research group [10,11].

Reduction of α- and β-unsaturated ketones **8** and **10** could lead to allylic alcohols **4** and **9,** respectively.

In Table 1, two different methods for the reduction of the α,β-unsaturated ketone **8** to the allylic alcohol **4** are presented. In both methods, the formation of a single alcohol with specific stereochemistry was observed. This indicates that regardless of the method used, the reaction with the α,β-unsaturated ketone **8** leads to the formation of a single stereoisomeric isomer of the allylic alcohol.

The reduction of enone **8** using L-Selectride in CH_2_Cl_2_ at −78 °C (Table 1, **entry 1**) furnished allylic alcohol **4** with a 56% yield. This reaction exhibited complete stereoselectivity towards the α face of the bicyclic compound. This is attributed to the steric hindrance imposed by the bulky nature of L-Selectride and 1,3-diaxial strain (A^1,3^) of the substrate, significantly hindering its access to the β face (Figure 3).

This organoboron compound demonstrates pronounced stereoselectivity by selectively attacking the α face of the carbonyl group. This preference for an axial attack of L-Selectride leads to forming a product with a specific stereochemical configuration.

On the other hand, the Luche reduction (Table 1, **entry 2**) is a specific process that uses lanthanides to carry out the reduction of unsaturated functional groups. This method yielded allylic alcohol **4** in a slightly higher yield of 60%. This type of reduction is a reaction that utilizes both a cerium-based reagent (such as CeCl_3_) and a borohydride reagent (such as NaBH_4_) in the same reaction to achieve stereoselective reduction of α,β-unsaturated ketones or aldehydes. This strategy combines the specific characteristics of both reagents to control the stereochemistry of the reaction. Lanthanoid addition allows a highly regioselective 1,2-reduction, which competes with the undesirable 1,4-reduction [12].

CeCl_3_ is a selective Lewis acid catalyst for the methanolysis of NaBH_4_. The resulting reagents, various sodium methoxyborohydrides, are harder reducing agents and therefore effect a 1,2-reduction with high selectivity. Furthermore, CeCl_3_ activates MeOH.

With allylic alcohol **4** in hand, the stage was now set for the synthesis of compound **5** (Figure 4) using an efficient method described by the Chugai group [13].

The reaction of alcohol **4** with *O*-phenyl chlorothionoformate resulted in the formation of carbonothioate **5** with a 72% yield, which underwent stereospecific [3,3]-sigmatropic rearrangement through the less hindered α face to afford a (20*S*)-thiaphenyloxycarbonyl derivative.

Under the same procedure, the alcohol 9 was converted to carbonothioate 5 with a 70% yield (Figure 5). Notably, under milder conditions, no conversion is observed, and it requires heating to 75 °C. This temperature is necessary to overcome the increased steric hindrance of the starting substrate [14].

The pericyclic reaction mechanism is illustrated in Figure 6, depicting the electronic rearrangement of π electrons within a cyclic transition state, leading to the formation of both σ and π bonds.

The reaction of **5** with Br(CH_2_)_2_CO_2_Et under alkaline conditions yielded methyl ester **6** in 60% yield and carboxylic acid **11** in 8% yield. Figure 7.

The rationale for the formation of ester **6** from carboxylic acid **11** can be explained using the Steglich esterification reaction [15]. Figure 8.

In Figure 9, we show a plausible mechanism involving the formation of ethyl ester **13** and subsequently hydrolysis and transesterification to afford **6** and **11**, respectively.

We were delighted to see that ester **6** could be recrystallized from a mixture of hexane and ethyl acetate (2:1) at −30 °C, and its structure was confirmed unambiguously as that shown in Figure 3, by X-ray crystallographic analysis [16].

## 3. Experimental Part

### 3.1. Materials and General Methods

All chemicals and dry solvents were purchased from ABCR, ACROS Organics, Alfa Aesar, Fisher Scientific, Fluorochem, Fluka, Merck, Sigma Aldrich, TCI, or VWR chemicals, and they were used as received without further purification unless stated otherwise. The addition of solvents and liquid reagents was performed via syringe or cannula. The Injekt syringes used were plastic (Braun) or Teflon (Hamilton) with Sterican needles. For low-temperature reactions, we used water/ice, CO_2_/acetone baths, or a Cryocool Haake EK90 Immersion Cooler.

All reactions involving non-aqueous solvents were carried out in flame-dried glassware sealed with rubber septa and under argon unless otherwise stated. Reactions were stirred using magnetic stirring bars, monitored by TLC performed on Merck Silica Gel 60 F254 aluminum sheets and visualized with UV (254 nm) fluorescence quenching and p-anisaldehyde stain (p-anisaldehyde (4.2 mL), AcOH (3.75 mL), conc. H_2_SO_4_ (12.5 mL) and MeOH (338 mL)). Concentrations under reduced pressure were performed by rotary evaporation at 40 °C at the appropriate pressure using Büchi R-II, R-200, and R-300. Chromatographic purification was performed as flash chromatography using silica gel Silicycle SiliaFlash^®^ P60 230–400 mesh under pressure. Unless otherwise stated, the yields given refer to chromatographically purified and spectroscopically pure compounds.

**NMR spectra** were recorded in CDCl_3_ at 298 K on Bruker ARX400 and AVANCE DPX400 spectrometers. All chemical shifts are reported in ppm with the residual solvent peak as the standard (CHCl_3_, δ_H_ = 7.26 ppm and δ_C_ = 77.2 ppm; CD_3_OH, δ_H_ = 4.87 ppm and δ_C_ = 49.0 ppm). The DEPT-135 pulse sequence was used to aid in the assignment of signals in the ^13^C NMR spectra. Peaks are reported as follows: δ (multiplicity (s = singlet, d = doublet, t = triplet, q = quartet, m = multiplet or unresolved, br = broad signal), coupling constant(s) in Hz, integration and assignment).

**MS analyses** were recorded on FTMS APEXIII and microTOF-Focus Mass Spectrometers by the mass spectrometry service of the Centro de Apoio Científico e Tecnolóxico á Investigación (CACTI) at the University of Vigo by Dr. Nieves Atanes and Dr. Manuel Marcos.

**IR spectra** were recorded on a Nicolet 6700 FT-IR spectrometer, equipped with an ATR using a Smart Orbit diamond crystal and/or a Nicolet Continuum Microscope by the Vigo Food Safety Service of CACTI at University of Vigo by Dr. Estefanía López.

**Melting point** (m.p.) was determined using the Stuart SMP10 melting point apparatus.

**HPLC analysis** was carried out on a Hewlett Packard 1050 system equipped with a DAD detector (254 nm), using an HPLC Phenomenex column (Luna 5μm Silica(2) 100 Å, 250 × 4.6 mm, isocratic mode; *^i^*PrOH/hexane). The analysis was performed by the Vigo Food Safety Service of CACTI at University of Vigo by Dr. Alberto Acuña.

**XRD analysis** was performed at 100 K using a Bruker D8 Venture diffractometer with a Photon II CMOS detector and Mo-Kα radiation by the Single Crystal X-Ray Diffraction Unit of CACTI at the University of Vigo by Dr. Berta Covelo.

### 3.2. Experimental Procedures

Synthesis of (8β)-(16β)-(17*E*)-Des-*A*,*B*-8,21-bis[(*tert*-butyldimethyl)silyloxy]pregn-17(20)-en-16-ol (**4**) (Figure 10).

To a solution of **8** (250 mg, 0.57 mmol, 1.0 equiv.) in MeOH (5.0 mL) at rt was added CeCl_3_·7H_2_O (339 mg, 0.91 mmol, 1.6 equiv.). The mixture was cooled to −78 °C, NaBH_4_ (35 mg, 0.91 mmol, 1.6 equiv.) was added, and the reaction was stirred at this temperature for 10 min. Then, the mixture was allowed to reach rt and was left standing for 19 h. The mixture reaction was quenched by the addition of H_2_O (5 mL), and the layers were separated. The aqueous phase was extracted with EtOAc (3 × 8 mL), and the combined organic extracts were dried over Na_2_SO_4_, filtered, and concentrated in vacuo. The crude material was purified by FCC (SiO_2_; using 2.5% EtOAc/hexane), affording compound **4** as a colorless oil (150 mg, 60%).

**Compound 4**: **^1^H-NMR** (400 MHz, CDCl_3_): δ 5.47 (td, *J* = 6.1, 1.7 Hz, 1H, H-20), 4.39 (ddd, *J* = 7.3, 6.7, 1.4 Hz, 2H, CH_2_-21), 4.28 (ddd, *J* = 13.3, 6.2, 1.6 Hz, 1H, H-16), 4.03 (c, *J* = 2.4 Hz, 1H, H-8), 2.05–1.96 (m, 1H, H-14), 1.95–1.81 (m, 2H), 1.77–1.62 (m, 3H), 1.51–1.32 (m, 4H), 1.25 (s, 3H, CH_3_-18), 0.90 (s, 9H, CH_3_-*^t^*Bu), 0.89 (s, 9H, CH_3_-*^t^*Bu), 0.07 (s, 6H, CH_3_-Si), 0.02 (s, 6H, CH_3_-Si); **^13^C-NMR** (101 MHz, CDCl_3_): δ 155.2 (C-17), 124.4 (CH-20), 74.7 (CH-8), 69.1 (CH-16), 59.6 (CH_2_-21), 48.3 (CH-14), 44.0 (C-13), 37.7 (CH_2_), 34.3 (CH_2_), 34.2 (CH_2_), 26.2 (CH_3_-*^t^*Bu), 26.0 (CH_3_-*^t^*Bu), 21.2 (CH_3_-18), 18.5 (C- ^*t*^Bu), 18.1 (C- ^*t*^Bu), 17.8 (CH_2_), −4.7 (CH_3_-Si), −4.9 (CH_3_-Si), −4.9 (CH_3_-Si), −5.0 (CH_3_-Si); **IR** (ATR, cm^−1^): ν 3360, 2950, 2929, 2856, 1471, 1254, 1080, 835, 773; **MS** (ESI): *m*/*z* (%) 463.3 ([M + Na]+, 4), 423.3 ([M-OH]+, 100); **HRMS** (ESI): *m*/*z* calculated for C_24_H_48_NaO_3_Si_2_ [M + Na]^+^ 463.3034, found463.3033; **TLC** (SiO_2_; 10% EtOAc/hexane): R*_f_* = 0.53. (Figure 4).

Synthesis of (8β)-(16β)-(17*E*)-Des-*A*,*B*-8,21-bis[(*tert*-butyldimethyl)silyloxy]pregn-17(20)-en-16-ol (**4**) (Figure 11).

To a solution of **8** (399 mg, 0.91 mmol, 1.0 equiv.) in anhydrous THF (3.0 mL) at −78 °C was added L-Selectride (1.0 M in THF, 2.7 mL, 2.70 mmol, 3.0 equiv.) dropwise, and the solution was stirred at this temperature for 16 h. The mixture reaction was quenched with sat. aq. NH_4_Cl solution (5 mL), and the layers were separated. The aqueous phase was extracted with EtOAc (3 × 5 mL), and the combined organic extracts were dried over Na_2_SO_4_, filtered, and concentrated in vacuo. The crude material was purified by FCC (SiO_2_; using 5% EtOAc/hexane), affording the product as a colorless oil (225 mg, 56%), whose physical constants and spectroscopic data virtually coincide with those described for this compound, using reduction with NaBH_4_.

Synthesis of Phenyl (8β)-(20*S*)-Des-*A*,*B*-8,21-bis[(*tert*-butyldimethyl)silyloxy]-24-nor-22-thiachol-16-en-23-oate (**5**) (Figure 12).

To a solution of **4** (150 mg, 0.34 mmol, 1.0 equiv.) in dry Py (2.2 mL) at rt was added DMAP (6.2 mg, 51 µmol, 15 mol%). The mixture was cooled to 0 °C, *O*-phenyl chlorothionoformate (95 µL, 0.68 mmol, 2.0 equiv.) was added, and the reaction was stirred at this temperature for 30 min. Then, the mixture was allowed to warm up to rt and was left standing for 16 h. The mixture reaction was quenched with H_2_O (5 mL), and the layers were separated. The aqueous phase was extracted with EtOAc (3 × 5 mL), and the combined organic extracts were washed with 15% *w*/*v* aq. CuSO_4_ solution (2 × 8 mL), dried over Na_2_SO_4_, filtered, and concentrated in vacuo. The crude material was purified (for purification prior to FCC, it is necessary to reduce the co-eluting impurity, *O*,*O*-diphenyl carbonothioate, using NaBH4 in MeOH at 0 °C.) by FCC (SiO_2_; using 0.5% EtOAc/hexane), affording compound **5** as a yellow oil (140 mg, 72%).

**Compound 5**: **^1^H-NMR** (400 MHz, Methanol-*d*_4_): δ 7.45–7.39 (m, 2H, H-3′, H-5′), 7.30–7.26 (m, 1H, H-4′), 7.16–7.12 (m, 2H, H-2’, H-6’), 5.77–5.73 (m, 1H, H-16), 4.18 (q, *J* = 2.6 Hz, 1H, H-8), 4.02–3.87 (m, 3H, H-20, CH_2_-21), 2.31 (ddt, *J* = 13.3, 11.7, 1.5 Hz, 1H, H-14), 2.04–1.88 (m, 2H), 1.76 (dtd, *J* = 11.9, 5.0, 3.6, 1.9 Hz, 3H), 1.66–1.52 (m, 3H), 1.11 (s, 3H, CH_3_-18), 0.93 (s, 9H, CH_3_-*^t^*Bu), 0.92 (s, 9H, CH_3_-*^t^*Bu), 0.11 (s, 3H, CH_3_-Si), 0.10 (s, 3H, CH_3_-Si), 0.08 (s, 3H, CH_3_-Si), 0.07 (s, 3H, CH_3_-Si); **^13^C-NMR** (101 MHz, CDCl_3_): δ 170.0 (C=O), 151.4 (C-17), 150.1 (C-1’), 129.6 (CH-3’, CH-5’), 127.1 (CH-4’), 126.2 (CH-16), 121.5 (CH-2’, CH-6’), 68.9 (CH-8), 65.2 (CH_2_-21), 54.6 (CH-14/20), 46.9 (C-13), 44.7 (CH-14/20), 35.0 (CH_2_), 34.7 (CH_2_), 31.3 (CH_2_), 26.0 (CH_3_-*^t^*Bu), 25.9 (CH_3_-*^t^*Bu), 18.9 (CH_3_-18), 18.4 (C-*^t^*Bu), 18.1 (C-*^t^*Bu), 18.0 (CH_2_), −4.7 (CH_3_-Si), −5.0 (CH_3_-Si), −5.2 (CH_3_-Si), −5.3 (CH_3_-Si); **IR** (ATR, cm^−1^): ν 2995, 2952, 2927, 2855, 1770, 1462, 1382, 1374, 1246, 1097, 1057, 927, 853, 837, 776; **MS** (ESI): *m*/*z* (%) 599.3 ([M + Na]^+^, 26), 577.3 ([M + H]^+^, 100), 445.2 ([M-OTBS]^+^, 25); **HRMS** (ESI): *m*/*z* calculated for C_31_H_52_NaO_4_SSi_2_ [M + Na]^+^ 599.3017, found 599.3017; **TLC** (SiO_2_; 10% EtOAc/hexane): R*_f_* = 0.86. (Figure 5).

Synthesis of Phenyl (8β)-(20*S*)-Des-*A*,*B*-8,21-bis[(*tert*-butyldimethyl)silyloxy]-24-nor-22-thiachol-16-en-23-oate (**5**) (Figure 13).

To a solution of **9** (820 mg, 1.86 mmol, 1.0 equiv.) in dry Py (10.0 mL) at rt was added DMAP (34 mg, 0.28 mmol, 15 mol%). The mixture was cooled to 0 °C, *O*-phenyl chlorothionoformate (1.0 mL, 7.44 mmol, 4.0 equiv.) was added, and the reaction was stirred at this temperature for 30 min. Then, the orange mixture was allowed to warm up to 75 °C and was left standing for 6 h. The mixture reaction was quenched with H_2_O (5 mL), and the layers were separated. The aqueous phase was extracted with EtOAc (3 × 5 mL), and the combined organic extracts were washed with 15% *w*/*v* aq. CuSO_4_ solution (2 × 8 mL), dried over Na_2_SO_4_, filtered, and concentrated in vacuo. The crude material was purified by FCC (SiO_2_; using 0.5% EtOAc/hexane), affording the product as a yellow oil (750 mg, 70%), whose physical constants and spectroscopic data virtually coincide with those described for this compound when the reaction was carried out at rt.

Synthesis of Methyl (8β)-(20*S*)-des-*A*,*B*-8,21-bis[(*tert*-butyldimethyl)silyloxy]-26,27-dinor-22-thiacholest-16-en-25-oate (**6**) and (8β)-(20*S*)-Des-*A*,*B*-8,21-bis[(*tert*-butyldimethyl)silyloxy]-26,27-dinor-22-thiacholest-16-en-25-oic acid (**11**) (Figure 14).

To a solution of **5** (300 mg, 0.52 mmol, 1.0 equiv.) in 10% *w*/*v* KOH/MeOH solution (4.2 mL) at rt was added ethyl 3-bromopropionate (660 µL, 5.19 mmol, 10.0 equiv.), and the resulting mixture was stirred at this temperature for 22 h. No progress was observed in the reaction with time, the mixture reaction was quenched by the addition of H_2_O (5 mL), and the layers were separated. The aqueous phase was extracted with EtOAc (3 × 8 mL), and the combined organic extracts were dried over Na_2_SO_4_, filtered, and concentrated in vacuo. The crude material was purified by FCC (SiO_2_; using 0.5–20% EtOAc/hexane), affording starting material **5** in the first fractions as a yellow oil (56 mg, 19%), ester **6** in the intermediate fractions as a colorless oil (165 mg, 60%), and carboxylic acid **11** in the last fractions as a colorless oil (23 mg, 8%).

**Compound 6**: **^1^H-NMR** (400 MHz, CDCl_3_): δ 5.48–5.41 (m, 1H, H-16), 4.10–4.04 (m, 1H, H-8), 3.86–3.80 (m, 1H, CH_2_-21), 3.79–3.74 (m, 1H, CH_2_-21), 3.66 (s, 3H, CH_3_-OMe), 3.19 (t, *J* = 6.9 Hz, 1H, H-20), 2.94–2.72 (m, 2H, CH_2_-23), 2.63–2.52 (m, 2H, CH_2_-24), 2.28–2.17 (m, 1H, H-14), 1.97–1.76 (m, 2H), 1.67 (dd, *J* = 11.6, 4.3 Hz, 3H), 1.61–1.40 (m, 3H), 1.00 (s, 3H, CH_3_-18), 0.86 (s, 18H, CH_3_-*^t^*Bu), 0.04 (s, 3H, CH_3_-Si), 0.03 (s, 3H, CH_3_-Si), 0.00 (s, 3H, CH_3_-Si), −0.00 (s, 3H, CH_3_-Si); **^13^C-NMR** (101 MHz, CDCl_3_): δ 172.6 (C=O), 151.4 (C-17), 124.5 (CH-16), 68.9 (CH-8), 67.5 (CH_2_-21), 54.5 (CH-14/20), 51.8 (CH_3_-OMe), 46.7 (C-13), 43.4 (CH-14/20), 35.1 (CH_2_), 35.1 (CH_2_), 34.7 (CH_2_), 31.1 (CH_2_), 26.7 (CH_2_), 26.0 (CH_3_-*^t^*Bu), 25.8 (CH_3_-*^t^*Bu), 18.9 (CH_3_-18), 18.4 (C-*^t^*Bu), 18.1 (C-*^t^*Bu), 18.0 (CH_2_), −4.7 (CH_3_-Si), −5.1 (CH_3_-Si), −5.2 (CH_3_-Si), −5.3 (CH_3_-Si); **IR** (ATR, cm^−1^): ν 2950, 2927, 2885, 2855, 1742, 1471, 1463, 1436, 1360, 1251, 1140, 1110, 1080, 1025, 1005, 972, 926, 833, 772; **MS** (ESI): *m*/*z* (%) 565.3 ([M + Na]^+^, 17), 543.3 ([M + H]^+^, 63), 411.2 ([M-OTBS]^+^, 100); **HRMS** (ESI): *m*/*z* calculated for C_28_H_55_O_4_SSi_2_ [M + H]^+^ 543.3354, found 543.3364; **TLC** (SiO_2_; 10% EtOAc/hexane): R*_f_* = 0.58. Figure 6.

**Compound 11**: **^1^H-NMR** (400 MHz, CDCl_3_): δ 5.52–5.41 (m, 1H, H-16), 4.13–4.07 (m, 1H, H-8), 3.88–3.80 (m, 1H, CH_2_-21), 3.84–3.76 (m, 1H, CH_2_-21), 3.21 (t, *J* = 6.9 Hz, 1H, H-20), 2.98–2.72 (m, 2H, CH_2_-23), 2.68–2.54 (m, 2H, CH_2_-24), 2.28–2.19 (m, 1H, H-14), 1.98–1.79 (m, 2H), 1.73–1.64 (m, 3H), 1.60 (dd, *J* = 12.7, 3.4 Hz, 3H), 1.56–1.41 (m, 2H), 1.01 (s, 3H, CH_3_-18), 0.88 (s, 18H, CH_3_-*^t^*Bu), 0.06 (s, 3H, CH_3_-Si), 0.05 (s, 3H, CH_3_-Si), 0.02 (s, 3H, CH_3_-Si), −0.01 (s, 3H, CH_3_-Si); **^13^C-NMR** (101 MHz, CDCl_3_): δ 178.4 (C=O), 151.3 (C-17), 124.7 (CH-16), 69.0 (CH-8), 67.6 (CH_2_-21), 54.5 (CH-14/20), 46.8 (C-13), 43.6 (CH-14/20), 35.2 (CH_2_), 34.7 (CH_2_), 31.2 (CH_2_), 26.4 (CH_2_), 26.0 (CH_3_-*^t^*Bu), 25.9 (CH_3_-*^t^*Bu), 19.0 (CH_3_-18), 18.5 (C-*^t^*Bu), 18.1 (C-*^t^*Bu), 18.0 (CH_2_), −4.7 (CH_3_-Si), −5.1 (CH_3_-Si), −5.2 (CH_3_-Si), −5.3 (CH_3_-Si); **IR** (ATR, cm^−1^): ν 2956, 2925, 2854, 1713, 1463, 1255, 1111, 1081, 1027, 836; **MS** (ESI): *m*/*z* (%) 591.2 ([M + NaCH_2_CN]^+^, 100), 551.3 ([M + Na]^+^, 9), 397.2 ([M-OTBS]^+^, 21); **HRMS** (ESI): *m*/*z* calculated for C_24_H_45_O_2_SSi [M + H]^+^ 529.3198, found 529.3197; **TLC** (SiO_2_; 10% EtOAc/hexane): R*_f_* = 0.18. Figure 7.

Synthesis of Methyl (8β)-(20*S*)-des-*A*,*B*-8,21-bis[(*tert*-butyldimethyl)silyloxy]-26,27-dinor-22-thiacholest-16-en-25-oate (**6**) (Figure 15).

To a solution of **11** (70 mg, 0.13 mmol, 1.0 equiv.) in dry CH_2_Cl_2_ (1.0 mL) at rt was added DMAP (2.4 mg, 19.5 µmol, 15 mol%) and DIC (24 µL, 0.16 mmol, 1.2 equiv.). The solution was stirred for 30 min and then added dry MeOH (38 µL, 0.66 mmol, 5.0 equiv.), and the mixture was left stirring for 28 h. The mixture reaction was diluted with Et_2_O (5 mL) and washed with sat. aq. NaHCO_3_ solution (2 × 5 mL) and brine (2 × 5 mL). The organic phase was dried over Na_2_SO_4_, filtered, and concentrated in vacuo. The crude material was purified by FCC (SiO_2_; using 1% EtOAc/hexane), affording the product as a colorless oil (64 mg, 90%), whose physical constants and spectroscopic data virtually coincide with those already described for this compound.

## 4. X-Ray Crystallography Data

X-ray crystallographic analysis is shown in Figure 3 and Appendix A.

## 5. Conclusions

In conclusion, we have demonstrated that by using our previously developed methodology, we could access a building block for synthesizing new thia analogs of calcitriol [17]. We have thus paved the way for easy access to a library of new analogs of TCT as well as their biological evaluation. Work is in progress for the synthesis of a series of such analogs.

## Data Availability

The data presented in this study are openly available in University of Vigo (https://biblioguias.uvigo.gal/guia-acordos-transformativos, accessed on 21 May 2025).

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
