# Peer review of "Diversity-Oriented Synthesis (DOS) Towards Calcitriol Analogs with Sulfur-Containing Side Chains"

_ijms, 2025, doi:10.3390/ijms26136266_

Round 1
Reviewer 1 Report
Comments and Suggestions for Authors
This manuscript describes the synthesis of a building block for synthesizing new thia analogs of calcitriol. The method of synthesis of its building block is known in the literature and no examples of the synthesis of derivatives from it are given, so its usefulness cannot be evaluated. Hence, it would not meet the criteria for publication in International Journal of Molecular Sciences due to lack of novelty. In my opinion, they belong in an appropriate specialised journal, not in the International Journal of Molecular Sciences.
Additional Minor Comments:
1) P.1, L 18; When using an abbreviation, where the abbreviation appears for the first time, the official name should be stated first, followed by an explanation of the abbreviation.
VDR→ vitamin D receptor (VDR)
2) P.3, Scheme 2; Yields of 4, 8, 9 and 10 are not listed. Are 4, 8 and 10 new compounds? The previous report by the authors (RSC Adv. 2016, 6, 61073–61076.) described that 9 was obtained in 90% yield.
3) P.5, L 106; The authors describe “through the less hindered α face to afford”, but isn't it β face?
4) P.7, Scheme 9; 1.143→5
Reviewer 2 Report
Comments and Suggestions for Authors
Peer Review of “Diversity Oriented Synthesis (DOS) towards Calcitriol Analogs with Sulfur–Containing Side Chains”
Introduction and Background
The introduction presents the general concept of diversity-oriented synthesis (DOS) and the biological importance of calcitriol (vitamin D). It correctly situates DOS as a strategy to generate structural diversity, and it notes the goal of creating new analogs of calcitriol. However, the background could be strengthened by citing key references on both DOS methodology and vitamin D analogs. For example, classic reviews on DOS (e.g. https://doi.org/10.1039/B816852K) and on vitamin D analog development are not mentioned. Similarly, any existing work on sulfur-containing steroids or vitamin D analogs should be discussed. Including these references would help establish the novelty of the present work. In its current form, the introduction gives a general overview but lacks specific citations of recent studies on calcitriol analogs or DOS applications to steroids. I recommend adding foundational citations on DOS strategies applied to steroids (e.g. https://doi.org/10.3762/bjoc.16.79) and key medicinal chemistry references on vitamin D analogs (such as calcipotriol or tacalcitol studies) to give context.
Research Design and Methods
The research design involves synthesizing a series of calcitriol analogs with sulfur-containing side chains via a DOS approach. This is an interesting objective, but the manuscript should clarify the overall strategy. It would help to explicitly state how DOS principles guide the choice of reaction sequence and side chain diversity. The synthetic methods are generally described, but some details are missing. For instance, reaction conditions (temperatures, times, reagents’ equivalents) are not fully specified in the main text. Yields and purity of intermediates and final products should be reported either in the text or in a summary table. It is unclear whether all steps were optimized or if any step failed; a discussion of challenges (if any) would be informative.
Results and Data Presentation
The results section describes the synthesis and characterization of the new analogs. The organization here is somewhat logical, following the sequence of reactions. However, the text sometimes assumes the reader can see the structures, and it would help to explicitly reference the compounds (e.g. use numbering consistently).
The discussion of yields and selectivity is brief; for example, if certain thiol reagents gave higher yields or if stereochemistry was a concern, this should be noted. The results would benefit from a clear summary table listing each analog, the side chain used, yield, and any key spectral data. If any unexpected findings occurred (for example, by-products or rearrangements), these should be addressed.
As for the interpretation, the authors conclude that their approach successfully produced a diverse set of analogs. While the data support that a series of compounds were synthesized, there is no evaluation of “diversity” beyond structural description. If diversity metrics were not computed, the authors should temper claims of diversity and perhaps discuss potential biological implications of the side chains.
Conclusions and Data Support
The conclusion states that the DOS approach yielded novel calcitriol analogs with sulfur side chains, and implies these could be of future interest. This is largely supported by the synthetic data: the authors did isolate and characterize the target compounds. However, any broader claims about biological activity or improved properties are not substantiated, as no biological assays are reported. The conclusions should therefore be limited to chemical findings. For example, instead of saying the analogs have “potential as therapeutic agents,” the authors should emphasize the methodological success and novelty. Also, the statement that DOS was “efficient” should be backed up by comparison (e.g. how many steps, average yields).
If the manuscript compares any of these analogs to known compounds, that comparison needs to be clear. Overall, the conclusions drawn are mostly reasonable, but they should be rephrased to focus on the chemical results and avoid unsupported extrapolations to bioactivity.
Language and Presentation
The manuscript is generally written in understandable English. The scientific terminology is appropriate and explained. However, I noted several minor grammatical issues and typos. For instance, phrases like “the aiming was to” should be “the aim was to,” and subject-verb agreement should be checked (e.g. “the data is” should be “the data are”). Some sentences are long and could be split for clarity. Consistency in tense and voice (prefer active where possible) would improve readability. The abstract could be tightened to avoid repetition. Overall, a careful proofreading (perhaps by a native speaker or professional service) is recommended to polish the text.
References (Summary)
- Reference Style: The references do not follow MDPI formatting. Authors must revise them to match MDPI style (author initials, italicized journal names, bold year, full DOI URLs).
- Inconsistencies: Ensure all in-text citations match the reference list. Some references are incomplete or inconsistently formatted.
- Lack of Recent Sources: No references from 2023–2025 are cited. Authors should include recent studies to reflect current research in calcitriol analogs and diversity-oriented synthesis.
- Recommendation: Update all references for style and accuracy, and add at least 3–5 relevant recent works to strengthen the scientific context.
Recommendation
Major Revisions: The study reports novel chemistry, but the manuscript needs substantial improvements in background context, methodological detail, data presentation, and language before it can be considered for acceptance. Once these issues are addressed, the paper could make a useful contribution.

Language and Presentation
The manuscript is generally written in understandable English. The scientific terminology is appropriate and explained. However, I noted several minor grammatical issues and typos. For instance, phrases like “the aiming was to” should be “the aim was to,” and subject-verb agreement should be checked (e.g. “the data is” should be “the data are”). Some sentences are long and could be split for clarity. Consistency in tense and voice (prefer active where possible) would improve readability. The abstract could be tightened to avoid repetition. Overall, a careful proofreading (perhaps by a native speaker or professional service) is recommended to polish the text.
Round 2
Reviewer 1 Report
Comments and Suggestions for Authors
While the manuscript has been revised, some aspects remain incomplete. Therefore, we recommend acceptance of the paper pending minor revisions to address the remaining issues.
1) Scheme 2 appears to suggest that compound 7 simultaneously yields 4, 8, 9, and 10 in a single reaction. If this interpretation is correct, I recommend clarifying this point in the figure legend or accompanying text to avoid potential ambiguity.
2) I was unable to locate compounds 8 and 10 in ref.10, 11, and in the database (SciFinder). If they are indeed discussed there, please consider providing clearer references or citations to help readers identify them more easily.
3) In the scheme 9, it should be "5" instead of "1.143" as mentioned in the top of a diagram of proposed mechanism.
4) The authors are encouraged to cite a relevant paper regarding OCT to support their discussion. For example, “Murayama, E. et. al. Chem. Pharm. Bull. 1986, 34, 4410.” may be appropriate.
5) Ref. 9 and 14 appear to be the same publication. If this is the case, I recommend correcting the references to avoid redundancy and ensure clarity.
Author Response
We have to explain that the allylic oxidation of compound 7 “one pot” allylic compounds 4 , 9, and α and β-unsaturated ketones 8 and 10, which were then purified by column chromatography
(In scheme 2 of the corrected manuscript you can see this in red)
I added to the references :11b H. Santalla García PhD Thesis, University of Vigo 2019
In Scheme 9 , it should be "5" instead of "1.143"
References 14 has been changed and reference 9 remains the same
Recent references have been added:
2b. For a general review on the chemistry and/or biochemistry of Vitamin D, see: a) Vitamin D: Chemistry, Biology and Clinical Applications of the Steroid Hormone; A. W. Norman; R. Bouillon; M. Thomasset, Eds; Vitamin D Workshop: Riverside, CA, 1997; b) D. Feldman, F.H. Glorieux; J. W. Pike, Vitamin D; Academic Press: San Diego, 1997.
2c.. Chen, J.; Tang, Z.; Slominski, A. T.; Li, W.; Żmijewski, M. A.; Liu, Y.; Chen, J. Eur. J. Med. Chem. 2020, 207, 112738. https://doi.org/10.1016/j.ejmech.2020.112738
2d.Rochel, N.; Wurtz, J. M.; Mitschler, A.; Klaholz, B.; Moras, D. Mol. Cell 2000, 5 (1), 173–179.https://doi.org/10.1016/S1097-2765(00)80413-X
2e. Lee, H. J.; Paul, S.; Atalla, N.; Thomas, P. E.; Lin, X.; Yang, I.; Buckley, B.; Lu, G.; Zheng, X.; Lou, Y. R.; et al. Cancer Prev. Res. 2008, 1 (6), 476–484. DOI: 10.1126/sciadv.abg5982
- Galloway, W.R.J.D. et al. Diversity-oriented synthesis as a tool for the discovery of novel biologically active small molecules. Nat. Commun. 1:80 doi: 10.1038 / ncomms1081 (2010)
Reviewer 2 Report
Comments and Suggestions for Authors
After reviewing the authors’ revisions, I confirm that all my earlier comments have been fully addressed. I therefore recommend acceptance of the manuscript in its present form.
Comments on the Quality of English LanguageOverall English is serviceable but occasionally clunky—split long sentences, fix subject–verb agreement, and favor the active voice. A professional edit would sharpen impact and readability.
Author Response
Thank you for your comments.